# COVID-19 Vaccine Hesitancy and Attitude toward Booster Doses among US Healthcare Workers

**DOI:** 10.3390/vaccines9111358

**Published:** 2021-11-19

**Authors:** Suman Pal, Rahul Shekhar, Saket Kottewar, Shubhra Upadhyay, Mriganka Singh, Dola Pathak, Devika Kapuria, Eileen Barrett, Abu Baker Sheikh

**Affiliations:** 1Department of Internal Medicine, University of New Mexico Health Sciences Center, Albuquerque, NM 87106, USA; spal@salud.unm.edu (S.P.); rshekhar@salud.unm.edu (R.S.); supadhyay@salud.unm.edu (S.U.); ebarrett@salud.unm.edu (E.B.); 2Department of Internal Medicine, Division of Hospital Medicine, University of Texas Health San Antonio, San Antonio, TX 78229, USA; kottewar@uthscsa.edu; 3Department of Medicine, Warren Alpert Medical School of Brown University, Providence, RI 02903, USA; mriganka_singh@brown.edu; 4Department of Statistics and Probability, Michigan State University, East Lansing, MI 48824, USA; dpathak@msu.edu; 5Division of Gastroenterology, Washington University School of Medicine in St. Louis, St. Louis, MO 63110, USA; devika.kapuria@gmail.com

**Keywords:** COVID-19, vaccine, booster, healthcare workers, United States

## Abstract

Vaccine reluctance among healthcare workers (HCW) can have widespread negative ramifications, including modeling behavior for the general population and challenges with maintaining a healthy workforce so we can respond to a resurgence of the pandemic. We previously reported that only one-third of HCW were willing to take the vaccine as soon as it became available prior to its Emergency Use Authorization (EUA). Here, we re-examine the attitude toward COVID-19 vaccines among HCW several months after the vaccines have been made widely available. In this study, only 7.9% (n = 107) of respondents were hesitant to take the first or second dose of the vaccine. Younger age (18–40 years) and lower level of education attainment (GED or less) were associated with higher vaccine hesitancy, whereas self-identified Asian racial identity was associated with greater acceptance of COVID-19 vaccination. Among the vaccine-hesitant group, more respondents noted mistrust of regulatory authorities (45.3%), government (48.6%), and pharmaceutical companies (50%) than mistrust of doctors (25.4%). Nearly two-thirds of respondents were concerned that vaccination may be ineffective against new strains and booster doses may be required; however, vaccine-hesitant respondents’ acceptance of a hypothetical booster dose was only 14.3%. Overall, vaccine hesitancy was observed to have demographic predictors similar to those previously reported; the hesitancy of some US HCW to receive booster doses may reflect a general hesitancy to receive other forms of vaccination.

## 1. Introduction

The World Health Organization (WHO) declared COVID-19, the disease caused by SARS-CoV-2, a pandemic on 11 March 2020 [1]. Since then, it has infected over 41 million individuals and led to more than 666,000 deaths in the US alone [2]. To contain the pandemic, several vaccine candidates were rapidly developed, rigorously tested, and made available for use. The Food and Drug Administration (FDA) granted an Emergency Use Authorization (EUA) to the first COVID-19 vaccine on 11 December 2020 [3]. Since then, vaccination efforts have been at the forefront of the measures to halt the pandemic and minimize further loss of lives. However, a low vaccine uptake can be a major barrier to containing the pandemic.

Healthcare workers (HCW) are a priority population for vaccination against COVID-19. As a high-risk population, HCW were prioritized as one of the first groups to receive the vaccine in keeping with recommendations by the Advisory Council on Immunization Practices [4]. HCW are also important sources of medical information and role modeling for the general population. Therefore, vaccine hesitancy among HCW can have widespread negative ramifications. We previously reported that, in the months preceding the announcement of EUA for Pfizer-BioNTech COVID-19 vaccine, only one-third of HCW in the US were willing to take the COVID-19 vaccine as soon as it became available, with a majority choosing to wait several months before deciding [5]. In this study, we examine the attitudes and hesitancy of HCW toward COVID-19 vaccines months after they were made available for use in this population. We also report on the current attitudes of HCW toward hypothetical additional booster doses of the vaccines.

## 2. Materials and Methods

Design: We conducted a cross-sectional survey to assess vaccine hesitancy among US HCW toward COVID-19 vaccination. An online English language survey tool was created using the REDCap electronic data capture tool hosted at the University of New Mexico. This survey was modified from our previously published study to gather more pertinent information regarding vaccine uptake among HCW. All responses were anonymized and no personally identifying information was collected. The study was conducted according to the guidelines of the Declaration of Helsinki and approved on 19 September 2020 by the Institutional Review Board of the University of New Mexico Health Sciences Center (study ID 20-515).

Sampling: The snowball sampling method was utilized. In this method of sampling, researchers initially reach out to their networks of contacts to recruit participants; agreeable participants may then recommend other potential participants from among their respective networks. This method of sampling was chosen for ease of recruitment with limited resources, as our populations of interest of HCW in the US were likely to be linked through professional contacts.

The survey tool was distributed via links emailed to employees of a large academic medical center in the southwestern US and posted to HCW groups on social media. Data were collected between 1 February 2021 and 31 March 2021.

Population: All adults (>18 years of age) working in a healthcare setting in the US, regardless of role or patient contact, were eligible to participate in the study. Informed consent was obtained prior to enrollment in the study. 

## 3. Outcomes

The main outcome of the study was identification of vaccine hesitancy among HCW with access to vaccination. Vaccine hesitancy was measured with a set of two questions. First, we asked respondents whether they had been offered any of the COVID-19 vaccines currently available in the US. Those who answered in the affirmative were then asked whether they received the vaccine(s). Their answers were captured as responses to one of the following questions: “Yes, I already took the first dose and plan to take the second dose/already taken the second dose”; “Yes I took the first dose but do not plan to take the second dose”; “No, I will wait to review real-world safety data”; “No I do not plan to take the vaccine”; and “Not sure”. Those who responded that they received both doses or were planning to receive both doses of the vaccine were identified as the vaccine non-hesitant group. Participants who did not accept either dose or responded that they were waiting or unsure were identified as the vaccine-hesitant group.

The primary analysis was performed by comparing these two groups (vaccine-hesitant and non-hesitant) to identify factors associated with vaccine hesitancy. The comparison was based on major demographic predictors: age, gender, race, ethnicity, education, income, occupation, primary area of work in healthcare (primary medical, primary surgical, etc.), type of healthcare facility (urban, rural, etc.), political identity, co-morbidity, self-perceived risk of contracting COVID-19, and experience of caring for patients with COVID-19. Based on occupation, the respondents were grouped as either direct medical providers (DMP) or direct patient-care providers (DPCP). DMP included physicians, resident physicians, medical students, and advanced-practice providers such as nurse practitioners (NPs) and physician assistants (PAs). DPCP included RNs, rehabilitation therapists, radiology technicians, nursing aides, social workers, care coordinators, administrative staff, and others without direct patient contact. 

Respondents’ attitudes toward COVID-19 vaccination in these groups, in terms of vaccine safety, efficacy, and trust, were assessed via their agreement to statements on a Likert scale. A descriptive analysis was performed by frequency distribution. 

A secondary outcome of the study was an assessment of the attitudes of HCW toward potential additional booster doses of COVID-19 vaccines. These attitudes were assessed with a set of two questions. First, we assessed the perception of need for booster doses by asking, “Are you worried that the current vaccine might not be effective against new strains, and you might require a booster dose or repeat vaccination?” We then assessed acceptance of booster doses with the question, “Would you be willing to take the vaccine/booster dose every year if required for sustained immunity?”.

## 4. Statistical Methods

Descriptive analyses were conducted on all study variables. Multiple logistic regression models were applied to test for plausible associations of vaccine hesitancy with age, gender, self-identified racial identity, ethnicity, highest level of educational attainment, income, occupation, primary area of work in healthcare (primary medical, primary surgical, etc.), type of healthcare facility (urban, rural etc.), history of taking influenza vaccine, co-morbidity, self-perceived risk of COVID-19 infection, and experience in taking care of patients with COVID-19. Each statistical test was conducted at α = 0.05, unless otherwise indicated. Data analysis was conducted using statistical software R (version 4.0.4).

## 5. Results

The survey was completed by 1374 respondents. Twelve of the respondents had not yet been offered the vaccine, and data on vaccine acceptance were unavailable for four respondents. Therefore, these respondents were excluded from the analysis.

Most of the respondents were between 30 and 60 years of age; female; white; held a bachelor’s degree or higher level of education; and identified as Democrat (Table 1).

GED: general educational development, indicating high school level of study; DMP: direct medical providers, including physicians, residents, fellows, medical students, NPs, PAs, and others; DPCP: direct patient care providers, including nurses, nursing aides, rehabilitation therapists, and others. 

A larger proportion of the respondents reported working in medical specialties as DMP, and in urban settings. Most participants reported no co-morbid conditions and perceived themselves to be at risk of slight-to-severe COVID-19 disease. A majority had directly taken care of COVID-19 patients. 

Of the 1358 participants who were offered the vaccine and whose full data were available, 107 (7.9%) were hesitant to receive either the first or second dose. Of those who were hesitant, 60 (56.1%) were waiting to review more data, 13 (12.1%) were not sure about their intention to vaccinate, and 31 (29.0%) did not plan to take the vaccine. Only three respondents (2.8%) were hesitant to take the second dose after receiving the first dose. All respondents who had received the first dose and were hesitant to take the second dose identified side effects with the first dose as the reason. 

Table 2 (and Appendix A) shows the characteristics of the two groups. In a logistic regression model for hesitancy, significant associations were noted for age, racial identity, education level, and political affiliation.

DMP: direct medical providers, including physicians, residents, medical students, NPs, PAs, and others. DPCP: direct patient care providers, including nurses, nursing aides, rehabilitation therapists, and others.

Younger age (18–30 years, OR 3.33, CI 1.48–8.23; 31–40 years, OR 2.74, CI 1.32–6.45) and lower level of education attainment (GED or lower, OR 2.739, CI 1.41–5.48) were associated with higher vaccine hesitancy. Self-identified Asian racial identity (OR 0.139, CI 0.02–0.63) was associated with lower levels of vaccine hesitancy.

Lack of trust was more common in the vaccine-hesitant group across all demographic domains (Figure 1).

Among the vaccine-hesitant group, lack of trust was reported in pharmaceutical companies (50%), government (48.6%), and regulatory authorities such as CDC/FDA (45.3%), with a much lower percentage reporting mistrust of doctors (25.4%). Those who were hesitant to receive COVID-19 vaccination noted concerns about vaccine safety (33.7%) and efficacy (19.8%) at higher rates than those in the non-hesitant group (concerns for safety −1.9%, concerns for efficacy−1.8%).

On the need for additional booster doses, most respondents (63.6%) were worried that current vaccination may not be effective against new strains and that additional booster doses or new vaccines may be needed for protection against emerging variants of the virus. These concerns were similar for the vaccine-hesitant (68.8%) and non-hesitant (63.1%) groups. When asked about acceptance of a hypothetical yearly booster vaccine to maintain immunity, overall acceptance was 83.6%. However, this acceptance was widely divergent among the two groups, with acceptance of a hypothetical annual booster dose at 13.8% among the vaccine-hesitant group and 89.9% among the non-hesitant group. (Figure 2A,B).

## 6. Discussion

Demand for COVID-19 vaccine has slowed in recent months [2,6,7,8,9,10], presenting a worrying trend that could delay achievement of herd immunity, allow for continued widespread circulation of the virus, and allow new variants to emerge and/or become more prevalent.

In the past year, hesitancy to receive vaccination against COVID-19 has been documented in the US and globally [11,12,13,14]. HCW were among the first subgroups of the population that were prioritized to receive the vaccine. Globally, COVID-19 vaccine hesitancy among HCW has been variable, ranging from <5% to >70% [15,16,17,18,19]. In the US, HCW acceptance of COVID-19 vaccine has also been varied. In our previous study conducted in October–November 2020, we reported that only approximately one-third of the surveyed population of HCW in the US were ready to take the vaccine as soon as it became available [5]. In subsequent studies, vaccine acceptance has been reported in various healthcare systems at 55.3–57.5% in December 2020, 86% in January 2021, and 84.6% in February 2021 [20,21,22,23]. This rising trend of vaccine acceptance is consistent with our findings. In the current study, we note that an overwhelming majority of respondents had either received or were willing to receive all recommended doses of the vaccine. Only 7.9% were unwilling to receive the vaccine, a finding that is similar to the findings of our previous study and lower than the vaccine refusal rates noted by Moniz et al. and Meyer et al. [5,20,21]. This difference could be due to differences among the studies in the framing of survey questions and the different time periods of the studies. 

Few demographic factors seem to be associated with vaccine hesitancy, although vaccine hesitancy is lower among higher-education-level and older-age HCW. These factors have been consistently noted in multiple studies of the general US population [11,14], as well as among HCW in the US [5,24,25] and globally [15]. Several factors may be responsible for the variation with age, including higher perceived risk of infection or serious illness, higher likelihood of presence of medical co-morbidities, and/or higher education, income, and experience in healthcare settings. 

Given the rise in cases of COVID-19 among young adults [26] and reports of complications among young adults [27], vaccine hesitancy could compound these problems. Accordingly, specific investigation into the causes of vaccine hesitancy is warranted so that mitigation strategies may be appropriately directed. Consistent with existing literature, our study confirmed that Asian racial identity was associated with lower vaccine hesitancy [5,15,24,25]. Although previous studies have noted higher vaccine hesitancy among Black HCW, our current study did not find a statistically significant difference in vaccine hesitancy between Black and White HCW. While this finding is encouraging, it may be due to the study being underpowered to detect any difference.

HCW who are hesitant to be vaccinated also state concerns about vaccine safety at much higher rates compared to non-hesitant HCW. While misinformation about adverse effects after COVID-19 vaccination has gained traction in segments of social and mass media, the robust monitoring systems under VAERS (vaccine adverse events reporting system) have shown very few serious adverse events causally linked to vaccination. Rates of serious adverse events such as anaphylaxis (2 to 5 people per million vaccinated), thrombosis with thrombocytopenia syndrome (46 reports after more than 14.5 million doses of Johnson & Johnson/Janssen vaccine and 2 after more than 362 million doses of mRNA COVID-19 vaccine), myocarditis, or pericarditis (854 confirmed reports after mRNA vaccine) are low. Death has been reported in 0.002% of people after COVID-19 vaccination, although a causal link with COVID-19 vaccine has not been established after review [28]. The data so far show that COVID-19 vaccines are safe; countering misinformation and disinformation will be key to reducing concerns about vaccine safety and increasing uptake.

HCW who are hesitant to be vaccinated have much higher levels of distrust in information provided by pharmaceutical companies, government, or regulatory authorities such as the CDC/FDA. Respondents who were hesitant to accept vaccination were more trusting of vaccination information provided by doctors and healthcare professionals than by these other sources of information; however, they were less trusting of doctors overall than were the respondents who were not hesitant to receive the COVID-19 vaccine. A perception of the beneficial role of healthcare providers has previously been suggested as a reason for vaccine uptake among children [29]. Thus, individual physicians and other healthcare professionals may be well suited to providing community messaging about vaccine acceptance.

Mounting evidence suggests a role for booster doses of COVID-19 vaccine in select populations. The FDA has issued EUA for the third dose of mRNA vaccines among solid organ transplant recipients and others who are similarly immunocompromised. Anticipating that, in the future, the scope for booster doses may broaden, we asked our respondents about their attitudes toward booster doses of COVID-19 vaccines. Their responses produced interesting findings. Nearly two-thirds of the respondents in each group (vaccine-hesitant and non-hesitant) were concerned that current vaccines may have decreased efficacy against new variants and booster doses may be required. However, when asked whether they would accept a yearly vaccine against COVID-19 to maintain immunity, acceptance was widely different between those who had taken the current vaccine and those who were hesitant to complete current vaccination. We hypothesize that hesitancy about initial COVID-19 vaccination may be a strong predictor for hesitancy about booster doses of the vaccine. This is unsurprising, as concerns over vaccine safety and mistrust of authorities, which were prevalent among hesitant HCW, are likely to extend to booster doses of the vaccine. Prior refusal of vaccination for other illnesses such as influenza has also been noted as a predictor of COVID-19 vaccine hesitancy [10,14]. It is reasonable to assume that this relationship would exist between refusal of initial COVID-19 vaccination and refusal of booster doses. However, it is interesting to note that a similar proportion of our respondents in both groups agree on the need for booster doses of the vaccine. This correlation could indicate that concerns beyond utility are the main drivers of reluctance to take booster vaccines, and may be an important consideration in the planning of messaging about booster doses. 

This study has limitations. As we utilized snowball sampling, our study population may not be representative of all US HCW, which limits the generalizability of our findings. The survey questionnaire was available only in an English online format; therefore, HCW with limited English proficiency and/or with limited internet access may be under-represented in our study. Social desirability bias may also affect the interpretation of our study, although the responses were anonymized to minimize this factor. 

## 7. Conclusions

The reluctance of some US HCW to receive COVID-19 vaccination is ongoing and is more likely to be prevalent among those of younger ages and lower levels of educational attainment. It is less likely among those with self-identified Asian racial identity. Concerns about vaccine safety, vaccine efficacy, and lack of trust were possible underlying causes of vaccine hesitancy. Physicians are more likely than pharmaceutical companies, regulatory authorities, or government to be trusted sources of information among HCW with COVID-19 vaccine hesitancy. Acceptance of COVID-19 vaccine booster doses may mirror a general hesitancy of some US HCW to receive other types of vaccination.

## Figures and Tables

**Figure 1 vaccines-09-01358-f001:**
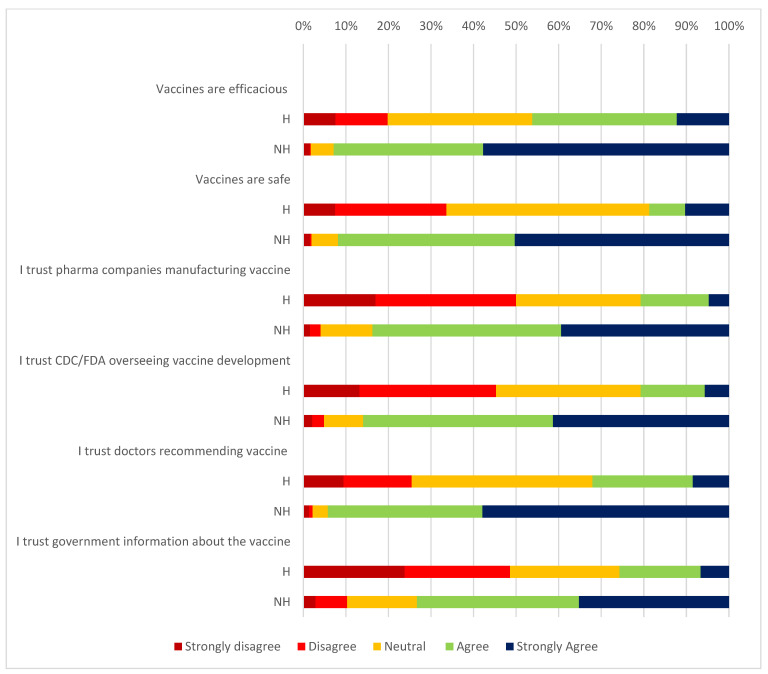
Attitudes related to COVID-19 vaccine. H—hesitant group, NH—non-hesitant group.

**Figure 2 vaccines-09-01358-f002:**
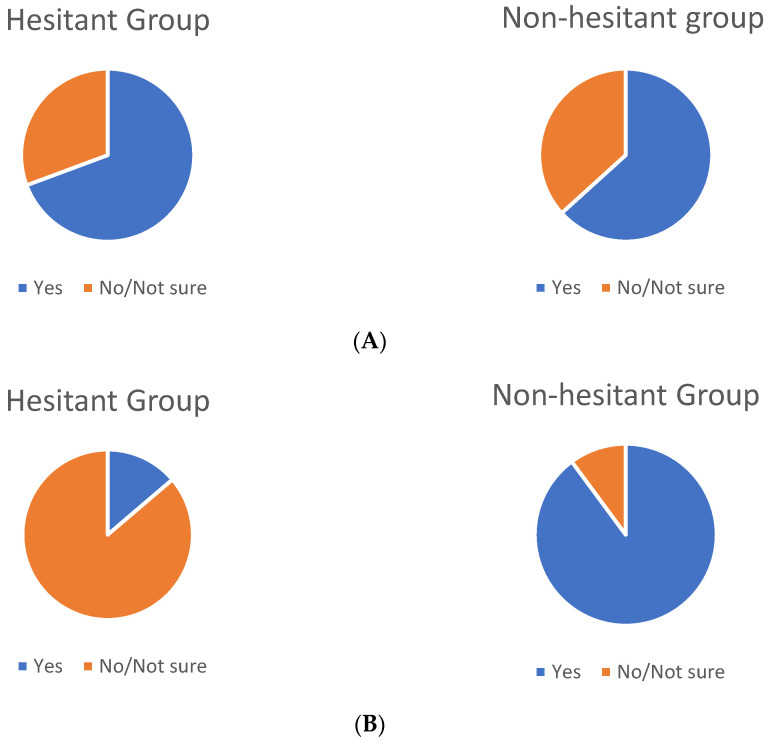
(**A**) Need for booster vaccine doses against new variants—“ Are you worried that the current vaccine might not be effective against new strains, and you might require a booster dose or repeat vaccination?” (**B**) Acceptance of potential yearly booster vaccine to maintain immunity—“Would you be willing to take the vaccine/booster dose every year if required for sustained immunity?”.

**Table 1 vaccines-09-01358-t001:** Demographics of study population.

Variable	Number (n = 1358)	Percentage
**Age ***		
18–30	170	12.5
31–40	384	28.3
41–50	292	21.5
51–60	294	21.6
>60	208	15.3
**Gender ***		
Female	1076	79.2
Male	273	20.1
**Racial identity ***		
Asian	91	6.7
Black	43	3.2
White	1132	83.4
Other	81	6.0
**Education**		
GED or less	168	12.4
Vocational	399	29.4
Bachelors	250	18.4
Masters	158	11.6
Professional	383	28.2
**Ethnicity**		
Hispanic/Latinx	134	9.9
**Variable**	**Number** (n = 1358)	**Percentage**
Not Hispanic/Latinx	1185	87.3
Unknown	39	2.9
**Occupation**		
DMP	542	39.9
DPCP	514	37.8
No direct contact	168	12.4
Administration	134	9.9
**Area of work**		
Medical	826	60.8
Surgical	114	8.4
Diagnostic	101	7.4
Other	317	23.3
**Type of facility**		
Urban	835	61.5
Suburban	437	32.2
Rural	86	6.3
**Co-morbidity**		
None	729	53.7
1	377	27.8
2	157	11.6
3 or more	95	7.0
**Variable**	**Number** (n = 1358)	**Percentage**
**Self-perceived risk**		
No, I already have recovered and won’t get re-infected (diagnosed by a test)	51	3.8
I believe I already had the disease and I am immune to it (not diagnosed by a test)	56	4.1
No, I am confident I won’t get infected	324	23.9
Yes, I am concerned that I will get infected	924	68.0
**COVID contact**		
Yes, direct contact	747	55.0
Yes, no direct contact	187	13.8
No	423	31.1

* Data missing so that the total of responses in these subgroups was <1358.

**Table 2 vaccines-09-01358-t002:** Comparison of hesitant and non-hesitant groups.

Variable	Hesitant Group (Number, percentage)	Non-hesitant Group (Number, percentage)
**Age ***		
18–30	20 (18.7)	150 (12.0)
31–40	38 (35.5)	346 (27.6)
41–50	23 (21.5)	269 (21.5)
51–60	18 (16.8)	276 (22.1)
>60	8 (7.5)	200 (16.0)
**Gender ***		
Female	91 (85.0)	985 (78.7)
Male	15 (14.0)	258 (20.6)
**Racial identity ***		
Asian	2 (1.9)	89 (7.1)
Black	6 (5.6)	37 (3.0)
White	90 (84.1)	1042 (83.3)
Other	9 (8.4)	72 (5.8)
**Education**		
GED or less	25 (23.4)	143 (11.4)
Vocational	39 (36.4)	360 (28.8)
Bachelors	15 (14.0)	235 (18.8)
Masters	12 (11.2)	146 (11.7)
Professional	16 (14.9)	367 (29.3)
**Variable**	**Hesitant group** (Number, percentage)	**Non-hesitant group** (number, percentage)
**Ethnicity**		
Hispanic/Latinx	12 (11.2)	122 (9.7)
Not Hispanic/Latinx	90 (84.1)	1095 (87.5)
Unknown	5 (4.7)	34 (2.7)
**Occupation**		
DMP	22 (20.6)	520 (41.6)
DPCP	64 (59.8)	450 (36.0)
No direct contact	6 (5.6)	162 (12.9)
Administration	15 (14.0)	119 (9.5)
**Area of work**		
Medical	66 (61.7)	760 (60.7)
Surgical	8 (7.5)	106 (8.5)
Diagnostic	11 (10.3)	90 (7.2)
Other	22 (20.6)	295 (23.6)
**Type of facility**		
Urban	61 (57.0)	774 (61.9)
Suburban	36 (33.6)	401 (32.0)
Rural	10 (9.3)	76 (6.1)
**Co-morbidity**		
None	58 (54.2)	671 (53.6)
1	25 (23.4)	352 (28.1)
2	14 (13.1)	143 (11.4)
**Variable**	**Hesitant group** (Number, percentage)	**Non-hesitant group** (number, percentage)
3 or more	10 (9.3)	85 (6.85)
**Self-perceived risk**		
No, I already have recovered and won’t get re-infected (diagnosed by a test)	7 (6.5)	44 (3.5)
I believe I already had the disease and I am immune to it (not diagnosed by a test)	19 (17.8)	37 (3.0)
No, I am confident I won’t get infected	15 (14.0)	309 (24.7)
Yes, I am concerned that I will get infected	65 (60.7)	859 (68.7)
**COVID patient contact**		
Yes, direct contact	56 (52.3)	691 (55.2)
Yes, no direct contact	19 (17.8)	168 (13.4)
No	32 (29.9)	391 (31.2)

* Missing data.

## Data Availability

Data will be made available on special request addressed to the corresponding author.

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
