# Peer review of "COVID-19 Vaccine Hesitancy and Attitude toward Booster Doses among US Healthcare Workers"

_vaccines, 2021, doi:10.3390/vaccines9111358_

Round 1

Reviewer 1 Report

The percentage data is not correct throughout in the tables. The abbreviations are not described like GED, NP, PA. EUA ect. Existing demographic predictors are not well described in abstract and conclusion.

Author Response

Dear Editors,

We have perused the reviewers’ comments and suggestions and have made changes to the manuscript to address them. Below are our responses (in italics) to each of the suggestions.

Reviewer 1

The percentage data is not correct throughout in the tables. The abbreviations are not described like GED, NP, PA. EUA ect. Existing demographic predictors are not well described in abstract and conclusion.

Response

We thank the reviewer for their suggestion. Abbreviations used have been expanded for better clarity.

We have also re-examined the percentages in table 1 and confirm that the percentages are accurate.

Reviewer 2 Report

You need to update your records in terms of current cases/deaths.

You should include the reported unwanted effects/ deaths related to vaccination in the introduction.

You clearly need to improve your tables and figure presentation.

Author Response

Dear Editors,

We have perused the reviewers’ comments and suggestions and have made changes to the manuscript to address them. Below are our responses (in italics) to each of the suggestions.

You need to update your records in terms of current cases/deaths.

You should include the reported unwanted effects/ deaths related to vaccination in the introduction.

You clearly need to improve your tables and figure presentation.

Response

We thank the reviewer for their suggestions.

The data on cases and deaths has been updated as of September 17, 2021.

We have added the following paragraph to our discussion ( instead of introduction as we feel it is a better fit in this section ) regarding adverse events /deaths related to vaccination.

HCWs who are hesitant to get vaccinated also note concern over vaccine safety at much higher rates compared to non-hesitant HCW. While misinformation of adverse effect after COVID-19 vaccine has gained traction in segments of social and mass media, the robust monitoring systems under VAERS (Vaccine adverse events reporting system) has shown very few serious adverse events causally linked to vaccination. Rates of serious adverse events such as anaphylaxis (2 to 5 people per million vaccinated), thrombosis with thrombocytopenia syndrome (TTS; 46 reports after more than 14.5 million doses of J&J/Janssen vaccine and 2 after mRNA covid vaccine after more than 362 million doses), myocarditis or pericarditis (854 confirmed reports after mRNA vaccine) are low. Death has been reported in 0.002% of people after the COVID vaccine though a causal link with the COVID-19 vaccine has not been established after review. [34] The data so far shows that COVID vaccines are safe and countering mis-/disinformation will be key to reducing concerns about vaccine safety and increase uptake.”

The tables and figures have been edited for improved clarity.

Reviewer 3 Report

Dear Authors, 

The paper titled "COVID-19 Vaccine Hesitancy and Attitude Toward Booster Doses Among US Healthcare Workers" seems very interesting and well written. However, in my opinion, considering the large population analyzed and the amount of the data collected, the results representation and the conclusions are very weak and it is not suitable for publication in the present form.

Major concern

  • Figure 1 should show the differences for other categories (such as: Area of work, occupation and so on). Indeed, the differences between hesitant and non-hesitant group do not clearly show these differences.
  • Figure 2A and 2B: The differences in hesitant group should be better explained in the discussion section. On this regard, it should be better if the Authors could hypothesize why such a big difference is present.
  • A better correlation of the data obtained should be performed to discuss the results. For example, the political orientation should be correlated with other parameters (such as: Education, occupation, area of work, occupation, type of facility).
  • The materials and methods section is too similar to previous published article [5].

Minor concern

- Please, specify in the tables legend the abbreviations "DPCP", "NP", "PA"

-Figure 1: It should be better to have a horizontal graph.

- Page 12, line 214: the reference number should be inside square brakets (...in our previous study [5]...).

Author Response

Dear Editors,

We have perused the reviewers’ comments and suggestions and have made changes to the manuscript to address them. Below are our responses (in italics) to each of the suggestions.

The paper titled "COVID-19 Vaccine Hesitancy and Attitude Toward Booster Doses Among US Healthcare Workers" seems very interesting and well written. However, in my opinion, considering the large population analyzed and the amount of the data collected, the results representation and the conclusions are very weak and it is not suitable for publication in the present form.

Response- We thank the reviewer for their suggestion. Specific concerns have been addressed as noted with each bullet point below.

Major concern

  • Figure 1 should show the differences for other categories (such as: Area of work, occupation and so on). Indeed, the differences between hesitant and non-hesitant group do not clearly show these differences.

Response – We emphasize that Figure 1 represents the attitude toward COVID vaccine among vaccine hesitant and non-hesitant group as measure by agreement to statements on a Likert scale. Since attitudes can be transient and modifiable, we chose to represent them separately.

However, we accept the reviewer’s suggestion that the variation in demographic characters of vaccine hesitant and non-hesitant group could be represented graphically and have included Figure 3 which may be added as supplement to the article.

  • Figure 2A and 2B: The differences in hesitant group should be better explained in the discussion section. On this regard, it should be better if the Authors could hypothesize why such a big difference is present.

Response – We thank the reviewer for this suggestion and interest in this finding. We have edited our discussion to further elaborate on these findings.

We hypothesize that hesitancy toward initial COVID-19 vaccination may be a strong predictor for hesitancy toward booster doses of the vaccine. This is unsurprising since concerns over vaccine safety and mistrust of authorities which were prevalent among hesitant HCWs, are likely to also extend to booster doses of the vaccine. Prior refusal of vaccination for other illnesses such as influenza has also been noted as a predictor of COVID-19 vaccine hesitancy [10,14]. It is reasonable to assume this relationship would exist between refusal of initial COVID-19 vaccination and booster doses. However, it is interesting to note that a similar proportion of our respondents in both groups agree to a need for booster doses of the vaccine. This could indicate that concerns beyond utility are the main drivers of reluctance to take booster vaccines. This would be important to consider when of messaging around booster doses.”

  • A better correlation of the data obtained should be performed to discuss the results. For example, the political orientation should be correlated with other parameters (such as: Education, occupation, area of work, occupation, type of facility).

Response – We would request further clarification of this suggestion. Does the reviewer recommend that we study association between various demographic factors carte blanche? In our opinion, this would be inappropriate as our sampling method does not produce a representative sample (as we note in our limitations) and sampling bias would make such associations difficult to interpret.

  • The materials and methods section is too similar to previous published article [5].

Response – We agree with this comment. However, since the study was designed as an extension of our prior survey, we have intentionally kept the survey instrument and method of sampling similar to our previous study therefore these sections are likely to read similar. The modifications done to the survey instrument are chiefly around how we identify the primary outcome as we elaborate in the outcomes section. Another addition to the survey were the questions on booster doses which we have also addressed in the outcome section.

Minor concern

- Please, specify in the tables legend the abbreviations "DPCP", "NP", "PA"

Response – This has been added to the footnote of the table.

-Figure 1: It should be better to have a horizontal graph.

Response – We accept the suggestion and have edited figure in a bar graph format.

- Page 12, line 214: the reference number should be inside square brakets (...in our previous study [5]...)

Response – This has been corrected.

Reviewer 4 Report

In this study, the authors pursue their previous work investigating COVID-19 vaccine attitudes and uptake among healthcare workers (HCWs) in the United States. In the present study, they re-examined attitudes towards COVID-19 vaccination, several months after vaccines became available. They found that only 8% of respondents were vaccine-hesitant (unwilling to receive a first or second dose, or unsure). However, they did find a high rate of concern about the possibility that vaccines may not be effective against new strains, or that repeated boosters may be necessary to maintain immunity.

Overall, the paper is well written, and the sample is large (although constituted by the snowball technique, which may induce some bias). The findings are interesting, and provide insights into the populations that need to be targeted for more focused interventions to achieve vaccine uptake, and also into the factors that characterize COVID-vaccine-hesitant individuals (in this study – age, ethnicity, education level and political leanings).

I have a few comments for the authors' consideration, in no specific order of importance:

  • In the abstract, the first sentence should be rephrased, because it is rather confusing: it reads “…. Negative ramifications, including…managing a workforce…”. How is managing a workforce a negative ramification of vaccine reluctance? The formulation needs to be changed here.
  • Line 30 (Abstract), the authors indicate that only 8% of respondents were hesitant, but they do not indicate the number of participants overall, or the actual number of those classified as vaccine hesitant. The numbers and percentages should be specified.
  • In Table 1, the authors should indicate the overall number of respondents at the top of the “Number” column.
  • Also in Table 1, the meaning of the different response modalities for “Self-perceived risk” is unclear – what is “no risk - will not”? Will not what? Will not contract COVID? Or will not get the vaccine? And what is “slight to severe”? Nobody can know in advance whether they would have a severe infection or not if infected. Either they feel themselves to be at risk (and the severity of infection is beyond their control) or not. I think you could just put “Yes” here, or “I consider myself at risk”.
  • Again, in Table 1, NP, PA are cited under the table but not defined. Either define or delete.
  • Second paragraph, page 6, the authors state that 1358 participants were offered the vaccine, and 107 were classed as hesitant – please indicate the percentage here. A breakdown of the responses prior to collapsing the categories into hesitant/non-hesitant would be useful (how many had one dose, how many refused outright, how many are unsure?)
  • In Table 2, the authors should indicate number AND percentage in both columns (with the totals in the column header). For gender, there appears to be one missing data (total for the hesitant group = 106??)
  • In Figure 2, it is unclear what the question was, and thus, correspondingly unclear what the results represent. In the “outcomes” section, the authors indicate that they asked respondents “whether they had concerns that current vaccine may be ineffective against new strains and a booster will be required”. Firstly, this could be reformulated, as it comprises two questions, and should read “whether they had concerns that current vaccines may be ineffective against new strains, and that a booster would be required”. Secondly, this does not quite correspond to what is indicated in the second paragraph of page 10, where the authors state “On the question of need for additional booster dose, most (63.6%) respondents reported that additional booster dose or new vaccines may be needed…”. This gives the impression that respondents were asked for their professional medical opinion about whether additional booster doses would be necessary to maintain immunity against COVID-19. The actual question needs to be clarified, so that it is clear what the respondents are answering about.
  • The authors should provide the survey as supplementary material.
  • Discussion, second paragraph, 13th line – the authors state that only 8% were unwilling to receive the vaccine. This is the first time you mention the percentage – this absolutely needs to be added to the results section (cf comments above).
  • Discussion, third paragraph – the authors indicate that “several factors may be responsible for this variation with age, including higher perceived risk… etc”. However, these factors appear to me to be more likely to explain higher uptake in older individuals, rather than lower uptake in younger individuals. The authors should phrase it the right way around, or else explain more clearly how higher perceived risk of infection or serious illness can explain why younger age is associated with higher hesitancy.
  • Discussion, fourth paragraph – the authors indicate that “our current study did not note a statistically significant difference in vaccine hesitancy among Black vs White HCW”. However, there are no data anywhere in the results section to support this statement, since the only ethnicity details provided were Latinx/Hispanic vs non Latinx/Hispanic. This sentence should be substantiated by including the appropriate results (or by changing the name of the non-Caucasian group to reflect inclusion of African-American respondents), or else removed.
  • Discussion, fifth paragraph – the second sentence needs to be corrected, and should read “This is consistent with findings….” (not “has been”). Also, in this same sentence, there is a stray reference that it unformatted.

Author Response

Dear Editors,

We have perused the reviewers’ comments and suggestions and have made changes to the manuscript to address them. Below are our responses (in italics) to each of the suggestions.

Overall, the paper is well written, and the sample is large (although constituted by the snowball technique, which may induce some bias). The findings are interesting, and provide insights into the populations that need to be targeted for more focused interventions to achieve vaccine uptake, and also into the factors that characterize COVID-vaccine-hesitant individuals (in this study – age, ethnicity, education level and political leanings).

Response – We thank the reviewer for their careful consideration of our paper and suggestions. These have been addressed as noted below with each specific comment.

I have a few comments for the authors' consideration, in no specific order of importance:

  • In the abstract, the first sentence should be rephrased, because it is rather confusing: it reads “…. Negative ramifications, including…managing a workforce…”. How is managing a workforce a negative ramification of vaccine reluctance? The formulation needs to be changed here.

Response – We accept the suggestion and have edited the sentence for more clarity. It now reads,
Vaccine reluctance among healthcare workers (HCW) can have widespread negative ramifications, including modeling behaviour for the general population and challenges with maintaining a healthy workforce so we can respond to a resurgence of the pandemic.”

  • Line 30 (Abstract), the authors indicate that only 8% of respondents were hesitant, but they do not indicate the number of participants overall, or the actual number of those classified as vaccine hesitant. The numbers and percentages should be specified.

Response – We agree with this suggestion and have added this data to the results section (Line 157).

  • In Table 1, the authors should indicate the overall number of respondents at the top of the “Number” column.

Response – This has been added to Table 1.

  • Also in Table 1, the meaning of the different response modalities for “Self-perceived risk” is unclear – what is “no risk - will not”? Will not what? Will not contract COVID? Or will not get the vaccine? And what is “slight to severe”? Nobody can know in advance whether they would have a severe infection or not if infected. Either they feel themselves to be at risk (and the severity of infection is beyond their control) or not. I think you could just put “Yes” here, or “I consider myself at risk”.

Response – We accept the suggestion and have edited the variable under self-perceived risk to better demonstrate the options provided to the respondents and improve clarity as well.

  • Again, in Table 1, NP, PA are cited under the table but not defined. Either define or delete.

Response – The abbreviations have been clarified in the footnote for the table.

  • Second paragraph, page 6, the authors state that 1358 participants were offered the vaccine, and 107 were classed as hesitant – please indicate the percentage here. A breakdown of the responses prior to collapsing the categories into hesitant/non-hesitant would be useful (how many had one dose, how many refused outright, how many are unsure?)

Response – Percentage has been added to line 147. The breakdown of the categories have also been added. The paragraph now reads,

“Of the 1358 participants who were offered the vaccine and whose full data was available, 107 (7.9%) were hesitant to receive either the first or second dose. Of those who were hesitant, 60 (56.1%) were waiting to review more data, 13 (12.1%) were not sure about their intention to vaccinate and 31(29.0%) do not plan to take the vaccine. Only 3 respondents (2.8%) were hesitant to take the second dose after receiving the first dose. All respondents who had received the first dose and were hesitant to take the second dose identified side effects with first dose as the reason.”

  • In Table 2, the authors should indicate number AND percentage in both columns (with the totals in the column header). For gender, there appears to be one missing data (total for the hesitant group = 106??)

Response – Percentages have been added to the columns in Table 2. Data is missing for some respondents in the variables of age, gender, and racial identity. This has been indicated in the footnote of the tables.

  • In Figure 2, it is unclear what the question was, and thus, correspondingly unclear what the results represent. In the “outcomes” section, the authors indicate that they asked respondents “whether they had concerns that current vaccine may be ineffective against new strains and a booster will be required”. Firstly, this could be reformulated, as it comprises two questions, and should read “whether they had concerns that current vaccines may be ineffective against new strains, and that a booster would be required”. Secondly, this does not quite correspond to what is indicated in the second paragraph of page 10, where the authors state “On the question of need for additional booster dose, most (63.6%) respondents reported that additional booster dose or new vaccines may be needed…”. This gives the impression that respondents were asked for their professional medical opinion about whether additional booster doses would be necessary to maintain immunity against COVID-19. The actual question needs to be clarified, so that it is clear what the respondents are answering about.

Response – The question has been added to the method section and figure legend for clarity. Results section has also been edited and now reads,

“On the question of need for additional booster dose, most (63.6%) respondents reported being worried that current vaccination may not be effective against new strains and that additional booster dose or new vaccines may be needed for protection against emerging variants of the virus.”

  • The authors should provide the survey as supplementary material.
  • Discussion, second paragraph, 13th line – the authors state that only 8% were unwilling to receive the vaccine. This is the first time you mention the percentage – this absolutely needs to be added to the results section (cf comments above).

Response – we have added this to the results section.

  • Discussion, third paragraph – the authors indicate that “several factors may be responsible for this variation with age, including higher perceived risk… etc”. However, these factors appear to me to be more likely to explain higher uptake in older individuals, rather than lower uptake in younger individuals. The authors should phrase it the right way around, or else explain more clearly how higher perceived risk of infection or serious illness can explain why younger age is associated with higher hesitancy.

Response – The sentence has been edited for clarity and now reads,

“Few demographic factors seem to be associated with vaccine hesitancy. Vaccine hesitancy is lower among higher education level and older age HCWs”

  • Discussion, fourth paragraph – the authors indicate that “our current study did not note a statistically significant difference in vaccine hesitancy among Black vs White HCW”. However, there are no data anywhere in the results section to support this statement, since the only ethnicity details provided were Latinx/Hispanic vs non Latinx/Hispanic. This sentence should be substantiated by including the appropriate results (or by changing the name of the non-Caucasian group to reflect inclusion of African-American respondents), or else removed.

Response – We have presented the data on racial identity and vaccine hesitancy in Table2.

  • Discussion, fifth paragraph – the second sentence needs to be corrected, and should read “This is consistent with findings….” (not “has been”). Also, in this same sentence, there is a stray reference that it unformatted.

Response – the sentence has been edited.

Reviewer 5 Report

Pal et al. presented findings on COVID19 vaccine hesitancy and openness to the third/booster dose conducted using an online survey on approximately 1300+ healthcare workers from southwest region of United States between 02/01/21 and 03/31/21. These findings are timely and highly relevant, and are appropriate for publication in Vaccines. Here are a few comments to improve on readability:

Tables 1 and 2: given that both tables span multiple pages, it is worthwhile to label the column headers for number and percentages for every sub-category.

Figure 1 suggestion - horizontal bar graph with groups on y-axis would significantly improve readability. 

Figure 2: border lines are unequal. perhaps remove lines altogether. Pie chart in different sizes.

Discussion and conclusion: By looking at the participation and survey findings, there are roughly similar numbers and percentages of participants that are hesitant in both Republican and Unaffiliated parties. However, authors only mentioned the former in both discussions and conclusions.

Author Response

Dear Editors,

We have perused the reviewers’ comments and suggestions and have made changes to the manuscript to address them. Below are our responses (in italics) to each of the suggestions.

Pal et al. presented findings on COVID19 vaccine hesitancy and openness to the third/booster dose conducted using an online survey on approximately 1300+ healthcare workers from southwest region of United States between 02/01/21 and 03/31/21. These findings are timely and highly relevant, and are appropriate for publication in Vaccines. Here are a few comments to improve on readability:

Response – We thank the reviewer for their comments and have accepted their suggestions for improvement. Specific responses are listed below.

Tables 1 and 2: given that both tables span multiple pages, it is worthwhile to label the column headers for number and percentages for every sub-category.

Response – We have added column headers for each new page for both tables.

Figure 1 suggestion - horizontal bar graph with groups on y-axis would significantly improve readability. 

Response – We agree with the suggestion and have edited figure 1 accordingly.

Figure 2: border lines are unequal. perhaps remove lines altogether. Pie chart in different sizes.

Response – We have removed borderlines and adjusted the size.

Discussion and conclusion: By looking at the participation and survey findings, there are roughly similar numbers and percentages of participants that are hesitant in both Republican and Unaffiliated parties. However, authors only mentioned the former in both discussions and conclusions.

Response – We accept the suggestion and have edited the sentences in both discussion and conclusion to reflect that Democrat political affiliation was associated with lower vaccine hesitancy.

Round 2

Reviewer 1 Report

There are a few "space" issues in typing. especially in abstract. Please make it clear that, what were your parameters to declared/ considered hesitant and non-hesitant. Please mention hesitant number in abstract. Please include in the conclusion why they were hesitant? Main possible identified reason (s) for hesitancy?.

Author Response

Dear Editors,

We have perused the reviewers’ and editor’s comments and suggestions and have made changes to the manuscript to address them. Below are our responses (in italics) to each of the suggestions.

There are a few "space" issues in typing. especially in abstract. Please make it clear that, what were your parameters to declared/ considered hesitant and non-hesitant. Please mention hesitant number in abstract. Please include in the conclusion why they were hesitant? Main possible identified reason (s) for hesitancy?

Response – We thank the reviewer for their suggestions. These have been addressed as noted below.

  • Space issues and grammar have been reviewed and errors resolved.
  • The identification of the two groups has been described in lines 95-108 and reads as.

The main outcome for the study was vaccine hesitancy among HCW with access to vaccination. This was measured with a set of two questions. We first asked respondents if they have been offered any of the COVID-19 vaccines currently available in the US. Those who answered in affirmative were then asked whether they received the vaccine(s) and responses were captured as “Yes, I already took the first dose and plan to take the second dose/ already taken the second dose”; “Yes I took the first dose but do not plan to take the second dose”; “No, I will wait to review real-world safety data”; “No I do not plan to take the vaccine”; and “Not sure”. Those who responded that they received both doses or were planning to receive both doses of the vaccine were identified as the vaccine non-hesitant group.  Participants who did not accept either dose or responded that they were unsure were identified as a vaccine-hesitant group.”

  • The number of hesitant respondents has been added to the abstract
  • The conclusion has been edited to reflect possible reasons for vaccine hesitancy.

Reviewer 3 Report

Dear Authors, 

The manuscript has been improved following the reviewers' suggestions. 

Just few typos are present such as at line 31, page 1, there is a need of a space ("only7.9%" should be written as: "only 7.9%"). Please fix it and read again, carefully, all the main text to avoid it.

Author Response

Dear Editors,

We have perused the reviewers’ and editor’s comments and suggestions and have made changes to the manuscript to address them. Below are our responses (in italics) to each of the suggestions.

The manuscript has been improved following the reviewers' suggestions. 

Just few typos are present such as at line 31, page 1, there is a need of a space ("only7.9%" should be written as: "only 7.9%"). Please fix it and read again, carefully, all the main text to avoid it.

Response – We thank the reviewer for their suggestions. The text has been reviewed carefully and errors of grammar and punctuation corrected.